# Prediction of Prognosis, Immunotherapy and Chemotherapy with an Immune-Related Risk Score Model for Endometrial Cancer

**DOI:** 10.3390/cancers15143673

**Published:** 2023-07-19

**Authors:** Wei Wei, Bo Ye, Zhenting Huang, Xiaoling Mu, Jing Qiao, Peng Zhao, Yuehang Jiang, Jingxian Wu, Xiaohui Zhan

**Affiliations:** 1Department of Bioinformatics, School of Basic Medical Sciences, Chongqing Medical University, Chongqing 400016, China; 2Department of Pathology, School of Basic Medical Sciences, Chongqing Medical University, Chongqing 400016, China; 3Department of Gynecology, The First Affiliated Hospital of Chongqing Medical University, Chongqing 400016, China; 4State Key Laboratory of Medical Genomics, Shanghai Institute of Hematology, National Research Center for Translational Medicine at Shanghai, Ruijin Hospital, Shanghai Jiao Tong University School of Medicine, Shanghai 200025, China; 5School of Basic Medical Sciences, Chongqing Medical University, Chongqing 400016, China; 6Department of Pathology, The First Affiliated Hospital of Chongqing Medical University, Chongqing 400016, China; 7Molecular Medicine Diagnostic and Testing Center, Chongqing Medical University, Chongqing 400016, China; 8Center for Medical Epigenetics, School of Basic Medical Sciences, Chongqing Medical University, Chongqing 400016, China

**Keywords:** endometrial cancer, co-expression network module, immune-related key genes, immune-based risk score model (IRSM), prognosis, immunotherapy, chemotherapy

## Abstract

**Simple Summary:**

Endometrial cancer (EC) is one of the most common gynecologic cancers. However, its clinical therapy remains unsatisfying due to the lack of effective treatment screening approaches. The primary treatment of EC is surgery, supplemented with radiotherapy and chemotherapy. In addition, immunotherapy as a promising therapeutic strategy has been gradually applied in clinical treatment. However, not all patients can benefit from such kind of clinical treatment, because EC is a heterogeneous disease and exhibits distinct patterns of molecular alterations, biological functions, as well as clinical outcomes. Thus, there is an urgent need to develop an effective model to help optimize treatment strategies and improve their therapeutic effects. Therefore, we aimed to construct a risk score model which could be used to predict the prognosis, immunotherapy response and chemotherapy sensitivity of EC. Our study provides insights into new personalized therapies and benefits EC treatment screening.

**Abstract:**

Endometrial cancer (EC) is the most common gynecologic cancer. The overall survival remains unsatisfying due to the lack of effective treatment screening approaches. Immunotherapy as a promising therapy has been applied for EC treatment, but still fails in many cases. Therefore, there is a strong need to optimize the screening approach for clinical treatment. In this study, we employed co-expression network (GCN) analysis to mine immune-related GCN modules and key genes and further constructed an immune-related risk score model (IRSM). The IRSM was proved effective as an independent predictor of poor prognosis. The roles of IRSM-related genes in EC were confirmed by IHC. The molecular basis, tumor immune microenvironment and clinical characteristics of the IRSM were revealed. Moreover, the IRSM effectiveness was associated with immunotherapy and chemotherapy. Patients in the low-risk group were more sensitive to immunotherapy and chemotherapy than those in the high-risk group. Interestingly, the patients responding to immunotherapy were also more sensitive to chemotherapy. Overall, we developed an IRSM which could be used to predict the prognosis, immunotherapy response and chemotherapy sensitivity of EC patients. Our analysis not only improves the treatment of EC but also offers targets for personalized therapeutic interventions.

## 1. Introduction

Endometrial cancer (EC) is the second most common gynecological cancer, with increasing incidence and mortality [1,2]. According to the global cancer statistics 2022, the incidence and mortality rate of EC are 3.44% (0.6595/19.18 million) and 2.06% (12,550/609,360), respectively [1]. In China, EC is also the most common gynecologic malignancy. The five-year survival rate of EC varies dramatically according to the Federation of Gynecology and Obstetrics (FIGO) stage at diagnosis. For EC patients with early- (FIGO stage I) and advanced-stage (FIGO stage IV) EC, the five-year survival rates are 90% and 15%, respectively [3,4]. Over the past decade, the overall survival of EC patients has improved but still remains unsatisfying due to the heterogeneity of EC and the lack of effective screening techniques. Therefore, to optimize treatment selection and improve patient outcomes, it is necessary to explore new effective strategies.

The main treatments of EC are surgery, neoadjuvant chemotherapy or radiotherapy and targeted therapy [5]. In addition, immunotherapy as a promising therapeutic strategy has been gradually applied in clinical treatment. However, only a proportion of patients receive clinical benefit from immunotherapy due to the fact that EC is a heterogeneous disease [6,7,8]. Thus, there is an urgent need to develop an effective model to optimize the treatment screening approach of EC. To date, numerous studies have been conducted to identify patients who might benefit from immunotherapy and chemotherapy [9,10,11,12,13,14]. Although remarkable achievements have been made, further studies are still required.

Gene co-expression network (GCN) analysis has been proven as an effective approach to elucidate the mechanisms and identify the drivers or drug targets of complex diseases [15,16]. Through GCN analysis, groups of genes presenting consistent expression patterns across specific disease conditions were mined, and these genes were often linked with specific biological functions. Currently, GCN analysis has been successfully applied to reveal the molecular basis as well as the drivers of various diseases [17,18,19].

In this study, we aimed to develop a reliable model to predict the survival of EC patients as well as their response to immunotherapy and chemotherapy. The workflow of this study is shown in Figure 1. Firstly, we applied the WGCNA algorithm to construct GCN modules and further mined the immune-related GCN modules. The yellow module which presented the closest correlation with the immune score was identified, and key genes of the yellow module were further obtained. Secondly, based on the above-identified immune-related key genes, we constructed an immune-related risk score model (IRSM) using the Lasso regression algorithm. The IRSM was found to be an independent poor prognosis predictor. Thirdly, we explored the molecular basis, tumor immune microenvironment and clinical characteristics of the IRSM. Meanwhile, we estimated the roles of key genes within the IRSM by IHC. We hypothesized that the IRSM effectiveness might be related to immunotherapy. Lastly, we further examined the relationship between the IRSM and immunotherapy and chemotherapy and highlighted the prediction ability of the IRSM for immunotherapy and chemotherapy. Meanwhile, we observed that immunotherapy coupled with chemotherapy could enhance the treatment effect. Overall, we established an immune-based risk score (IRSM) model which could be used to predict patient prognosis and response to immunotherapy and chemotherapy. 

## 2. Materials and Methods 

### 2.1. Data Acquisition and Preprocessing

Two endometrial cancer (EC) datasets including the TCGA-UCEC dataset and The First Affiliated Hospital of Chongqing Medical University dataset were used in our analysis. For the former (TCGA-UCEC) dataset, matched gene expression data (counts data), somatic mutation data and clinical data were used in this analysis. Both gene expression data and mutation data were obtained from the GDC TCGA Data Portal (https://portal.gdc.cancer.gov/, accessed on 13 January 2022). As for the clinical data, they were downloaded from the GDC TCGA Data Portal and the “TCGABiolinks” R package [20]. For the gene expression data, only mRNAs were extracted according to the Gencode GTF annotation file, and log2(count + 1) transformations were performed [21]. Then, the median absolute deviation (MAD) for each gene was calculated, and the top 5000 most variable genes were chosen for further analysis. For the clinical data, patients with missing overall survival information were excluded. For the latter (The First Affiliated Hospital of Chongqing Medical University) dataset, 20 EC tissues and 7 matched normal tissues were used for immunohistochemical (IHC) analysis. The demographic and clinical characteristics of the patients are described in Table 1.

### 2.2. Construction of the Weighted Gene Co-Expression Network 

To identify gene co-expression network (GCN) modules, we applied the weighted gene co-expression network analysis (WGCNA) tool to mine the GCN modules [15]. Firstly, we clustered the samples to check whether there were outlier samples based on hierarchical clustering analysis. We applied the goodSamplesGenes function to check if there were genes and samples with too many missing values. We observed that all the genes and samples passed the examination. After that, we performed hierarchical clustering of the samples by the hclust function to detect if there were any obvious outliers. A clustering dendrogram of the samples based on Euclidean distance was constructed, and no outlier samples were generated (Appendix A). Secondly, we screened the most optimal soft threshold β to construct a scale-free network. Thirdly, we constructed an adjacency matrix based on the Pearson correlation coefficient (r), which was subsequently converted into a topological overlap matrix (TOM). Fourthly, we mined the GCN modules using the Dynamic Branch Cut method. Especially, the minimal module size and the cutting height for module merging were set to 30 and 0.25, respectively. Finally, we calculated the module eigengenes (MEs) for each module, which was the first principal component of the GCN module and could represent the overall expression level of the module.

### 2.3. Identification of Immune-Related Key Genes

To identify immune-related key genes, we first explored the immune-related GCN modules. For each GCN module, the correlation between the MEs and the immune characteristics (immune score calculated by the ESTIMATE algorithm) was quantified based on the Pearson correlation coefficient (r), and the most correlated GCN module was defined as the immune-related GCN module [22]. 

Next, we explored the key genes of the immune-related GCN module. Gene significance (GS) and module membership (MM) were initially adopted to assess the relationship between the module genes and the immune characteristics. GS represents the association of module genes with the immune score. MM represents the association of module eigengenes (MEs) with module genes. The potential hub genes were initially screened with GS > 0.8 and MM > 0.8. Meanwhile, we also screened the potential hub genes with Protein–Protein Interaction (PPI) Network analysis [23], and the top 50 genes with a high node degree were considered. The genes at the intersection of the above two kinds of potential hub genes were considered immune-related key genes and used for further analysis.

### 2.4. Estimation of the Immune Score

The ESTIMATE algorithm is a powerful unsupervised approach to explore the tumor microenvironment [22]. It consists of three core modules including the immune score module, the stromal score module and the ESTIMATE score module. Based on the immune score module, the relative abundance of infiltrating immune cells in tumor tissue could be measured according to the gene expression profile. Particularly, the immune score has been considered as an effective biomarker to estimate the intratumoral immune status and predict the patient survival and response to immunotherapy.

### 2.5. Functional Enrichment Analysis

To explore the biological function of the immune-related module, we performed Gene Ontology (GO) enrichment analysis based on the module genes with the “clusterProfiler” R package [24]. The hypergeometric test was used to calculate the *p*-value for gene set enrichment, and the false discovery rate (BH FDR, Benjamini–Hochberg False Discovery Rate) was used to calculate the q value for multiple test compensation. Only biological processes with a q-value less than 0.05 were considered significantly enriched, and the top 10 enriched biological processes were displayed using the “Graphics” R package [25].

### 2.6. Construction and Validation of the Risk Score Model

To construct a risk score model, for each previously identified key gene, we first applied univariate Cox regression analysis to evaluate the relationship with patient survival outcome by the “survival” R package [26]. Genes with *p* values less than 0.05 were considered as immune-related prognosis features and were used to construct the model. Next, we randomly divided the patients into training and validation datasets (7:3). To make a more practical model, we normalized the mRNA data and adopted the Lasso regression algorithm by the “glmnet” R package [27] to select the most immune-related prognosis genes. Then, we constructed an immune-related risk score model (IRSM) with the above selected genes in the training dataset. For this IRSM:Risk score = Σ(coef (G) × EXP(G))

G: immune-related prognosis gene signature; coef: Lasso cox regression coefficient; EXP(G): expression value of the gene signatures.

Based on the IRSM, each patient was assigned a risk score, and the risk score was the weighted combination of the Lasso Cox regression coefficients and the corresponding expression value of the gene signature. Using the median risk score as the cutoff, we divided the patients within the training dataset into high- and low-risk groups. Then, we used the Kaplan–Meier estimator [28] to estimate the survival time, and the log-rank test was applied to compare the survival difference between the high- and the low-risk groups. Using a similar method, we tested the prognosis prediction performance of the IRSM in the validation dataset and the whole dataset separately.

### 2.7. Validation of the Risk Score Model by Immunohistochemistry

To validate the efficacy of the IRSM for EC tumorigenesis and development, we applied immunohistochemistry (IHC) staining to study the five genes in the risk score model. Twenty EC tissues and seven normal tissues collected from The First Affiliated Hospital of Chongqing Medical University were included, and protein expression from each gene was compared. The main steps for IHC are listed below: (1) all tissues were fixed with 4% neutral formaldehyde, embedded in paraffin wax and then cut into 4 μm thick sections; (2) the sections were baked at 60 °C for 1 h and then dewaxed using xylene (twice, 10 min each time) and rehydrated through a gradient alcohol series; (3) after being washed, the sections were immersed in a hydrogen peroxide solution for 8 min and in a PBS solution thrice (5 min each time). Subsequently, the sections were boiled in a dilute sodium citrate antigen retrieval solution for 5 min for antigen retrieval; (4) the sections were incubated with the primary antibodies overnight at 4 °C and with the secondary antibodies for 30 min at 37 °C. The primary antibodies were: anti-CD3D primary antibody (1:100; SAB, Johannesburg, South Africa; 38225), anti-CD3E primary antibody (1:100; ABClonal, Woburn, MA, USA; A19017), anti-CXCR3 primary antibody (1:100; BOSTER, Pleasanton, CA, USA; PB9079), anti-CCL5 primary antibody (1:100; Abbkine, Wuhan, China; ABP56130) and anti-IL-2RG primary antibody (1:500; Abcam, Shanghai, China; AB273023); (5) after DAB staining and washing, hematoxylin was used to counterstain the samples, and a hydrochloric acid solution was used to dissociate the samples; (6) lithium carbonate in saturated aqueous solution was employed to promote blue color development; (7) the expression level of each protein was estimated by microscopy, and representative images were captured.

### 2.8. The Molecular Basis of the Risk Score Model

To explore the molecular basis of the IRSM, we compared the expression pattern of IRSM-related genes as well as yellow module genes in the high- and low-risk groups by the Wilcoxon test (*p* < 0.05 indicated significant differences).

### 2.9. Relationship of the Risk Score Model with Clinical and Molecular Characteristics

To assess the relationship between the IRSM with clinical as well as molecular characteristics, we first performed multivariate Cox analysis to assess whether the IRSM was independent of age, FIGO grade, FIGO stage and pathological subtype.

Considering the important roles of clinical and molecular characteristics such as microsatellite instability (MSI), *POLE* and *TP53* mutation, mismatch repair gene (MMR, including *MLH1*, *MSH2*, *MSH6* and *PMS2*), immune checkpoint genes (*CD274*, *CTLA4*, *HAVCR2*, *LAG3*, *PDCD1*, *PDCD1LG2*, *TIGIT* and *TNFRSF18*) and human leukocyte antigen (HLA) genes (*HLA-A*, *HLA-B*, *HLA-C*, *HLA-DMA*, *HLA-DMB*, *HLA-DOA*, *HLA-DOB*, *HLA-DPA1*, *HLA-DPB1*, *HLA-DQA1*, *HLA-DQA2*, *HLA-DQB1*, *HLA-DQB2*, *HLA-DRA*, *HLA-DRB1*, *HLA-DRB5*, *HLA-E*, *HLA-F* and *HLA-G*) in EC tumorigenesis and development, we also compared these clinical and molecular characteristics in patients in the high- and low-risk groups. The Fisher exact test was used to compare the MSI-H status, the mutation frequency of MMR, *TP53* and *POLE* between high- and low-risk groups (*p* < 0.05 was considered to indicate a significant difference). The visualization of mutations in the above genes was performed by the “maftools” R package [29]. In addition, the Wilcoxon test was used to compare the expression patterns of immune checkpoint genes and HLA genes between the high- and the low-risk groups (*p* < 0.05 was considered to indicate a significant difference).

The Tumor Mutational Burden (TMB) is considered a promising biomarker for immunotherapy and is associated with a better immunotherapy response [30,31,32]. To better assess the relationship between the IRSM and immunotherapy, we firstly calculated the TMB for each patient by the “maftools” R package [29]. A relatively strict and widely used cutoff of TMB (top 20% of TMB) was adopted to estimate the TMB status [30,33]. A high TMB (TMB-H) indicated that the TMB was above the cutoff, while a low TMB (TMB-L) indicated that the TMB was below the cutoff. The Fisher exact test was then applied to compare the differences between the two groups (*p* < 0.05 was considered to indicate a significant difference).

### 2.10. Estimation of Immune Infiltrating Cell Contents

To estimate the immune infiltrating cell contents within the tumor environment, we first evaluated the immune cell proportion in each patient by calculating the immune score with the ESTIMATE algorithm [22], which was also used to identify the immune-related GCN modules. Moreover, we also measured the relative distribution of 22 types of infiltrating immune cells with the CIBERSORT algorithm [34,35]. The abundance of the 22 types of infiltrating immune cells was subsequently compared in the high- and low-risk groups using the Wilcoxon test (*p* < 0.05 was considered to indicate a significant difference).

### 2.11. Relationship between the Risk Score Model and Immunotherapy

Tumor immune dysfunction and exclusion (TIDE) is a computational method developed by X. Shirley Liu et al. to predict the immune checkpoint blockade (ICB) response [36,37]. Currently, this approach has been successfully applied to multiple cancer studies including non-small cell lung cancer and melanoma to predict the ICB response [38,39,40]. Thus, the TIDE is considered a powerful tool to predict the ICB response. To evaluate the relationship between the IRSM and immunotherapy, we first predicted the ICB response according to the TIDE website (http://tide.dfci.harvard.edu, accessed on 14 April 2022) [36,37]. The TIDE score for each patient was obtained. A high TIDE score indicates a high potential of tumor immune evasion, which means the patient is less likely to benefit from an anti-PDL1/CTLA4 treatment. Next, we compared the TIDE scores of the patients in the high- and low-risk groups using the Wilcoxon test (*p* < 0.05 was considered to indicate a significant difference). 

### 2.12. Relationship between the Risk Score Model and Chemotherapy

To evaluate the relationship between the IRSM and chemotherapy, we first computed the half-maximal inhibitory concentration (IC50) of 7 commonly used chemotherapeutic drugs (Cisplatin, Cyclophosphamide, Docetaxel, Fluorouracil, Paclitaxel, Tamoxifen and Topotecan) using the “oncoPredict” R package [41]. The IC50 is widely used as a measure of drug effectiveness, and patients with a high IC50 value are less sensitive to the corresponding drug. Next, we compared the IC50 of the patients in the high- and low-risk groups with the Wilcoxon test (*p* < 0.05 was considered to indicate a significant difference). 

### 2.13. Statistical Analysis Software

Except where noted above, all statistical analyses were performed in R version 4.1.2.

## 3. Results

### 3.1. Screening of Immune-Related Key Genes

Gene co-expression network (GCN) analysis has been proven as an effective approach for mining potential drivers or drug targets. We performed GCN analysis with the weighted gene co-expression network analysis (WGCNA) algorithm to identify immune-related modules. After pre-processing the gene expression data, 5000 genes with the most variable expression values across the samples were used to construct GCN modules. Particularly, the optimal soft-thresholding β = 4 (scale-free R^2^ = 0.9) was employed to construct a scale-free network (Figure 2A), Pearson correlation coefficient (r) was used to construct an adjacency matrix, and the dynamic tree cut algorithm was adopted to generate the GCN modules. As a result, eight consensus GCN modules were obtained for EC (Figure 2B). Meanwhile, for each module, the module eigengenes (MEs) which could represent the whole expression level of the modules were also calculated.

To mine immune-related GCN modules, we adopted the Pearson correlation coefficient (r) to estimate the relationship of previously identified GCN modules with immune traits. Here, the immune score calculated from the ESTIMATE algorithm was used to represent the immune traits of each patient. Specifically, the yellow module (377 genes) which exhibited the strongest positive correlation (r = 0.78, *p*-value = 5 × 10^−118^) with the immune score stood out (Figure 2C). In addition, we performed GO enrichment analysis to explore the biological processes of the yellow module. We observed that immune-related biological processes such as T cell activation (q-value = 3.129329 × 10^−53^) and regulation of T cell activation (q-value = 8.881909 × 10^−43^) were significantly enriched (Figure 2E), indicating that the yellow module was indeed immune-related.

To further mine immune-related key genes, we explored the potential hub genes of the yellow module with two algorithms. Firstly, after screening with Gene Significance (GS) for module genes with immune score GS > 0.8 and module membership (MM) in the yellow module with MM > 0.8, 11 potential hub genes of the yellow module were identified (Figure 2D). Next, Protein–Protein Interaction (PPI) Network analysis of the yellow module was performed, and the top 50 genes with a high node degree were considered as potential hub genes (Appendix A). Subsequently, the potential hub genes obtained from the above methods were intersected, and seven immune-related key genes (*CXCR3*, *CD3D*, *CD2*, *CCL5*, *CD3E*, *IL2RG* and *ITGAL*) were identified. Taken together, seven immune-related key genes were mined with the WGCNA algorithm.

### 3.2. Construction of the IRSM Based on Immune-Related Key Genes

Given the weakening of the immune function during cancer progression, we firstly estimated whether the immune-related key genes held clinical survival information. For each immune-related key gene, univariate Cox regression analysis was applied to estimate its relationship with the survival outcome. We observed that all these immune-related key genes were survival-associated (*p*-value < 0.05), and the high expression of these genes was linked to good prognosis (Appendix A). Then, all the TCGA-UCEC patients were randomized into a training and a testing group in the ratio of 7:3. For the training dataset, we constructed an immune-related risk score model (IRSM) using the Lasso regression algorithm (Appendix A). For the IRSM, five key genes (*CXCR3*, *CD3D*, *CCL5*, *CD3E* and *IL2RG*) were identified using a weighted combination of the Lasso cox regression coefficients and their corresponding expression value. In this IRSM:Risk score = (−0.2) × Expression (*CXCR3*) + (0.3) × Expression (*CD3D*) + (0.3) × Expression (*CCL5*) + (−0.01) × Expression (*CD3E*) + (0.1) × Expression (*IL2RG*)

### 3.3. The IRSM Could Serve as a Prognosis Predictor of EC

According to the IRSM, each patient was assigned a risk score. To estimate the relationship of the IRSM with each patient’s clinical outcome, all the patients in the training dataset were categorized into two groups (high- and low-risk groups) based on the median risk score. Compared with the low-risk group, the patients in the high-risk group had a shorter survival time (log-rank *p*-value = 0.0017, Figure 3A), suggesting that the IRSM is associated with poor prognosis.

To test whether the use of the IRSM as an EC survival predictor over-fitted the data, similar analyses were conducted on the validation dataset and the whole dataset to estimate the predictive performance of the IRSM. We observed that in both the validation dataset and the whole dataset, the patients in the high-risk group had a significantly shorter survival time than the patients in the low-risk group (Figure 3B,C). This phenomenon was consistent with the training dataset and confirmed the robustness of the IRSM for prognosis prediction. In short, the IRSM could serve as a poor-prognosis predictor for EC patients.

### 3.4. IHC Confirmed the Effect of IRSM-Related Genes on EC

To validate the role of the genes associated with the IRSM in EC development and progression, 20 EC tissues and 7 normal tissues were used, and immunohistochemistry (IHC) was applied to compare the corresponding protein expression levels. IHC staining showed stronger expression of CD3D, CD3E, CXCR3 and IL2RG in tumor tissue compared to normal tissue. As for CCL5, strong or moderate staining in tumor tissue and negative staining in normal tissue were noticed (Figure 4). All of these results confirmed the important roles of the IRSM-related genes in EC.

### 3.5. The Molecular Basis of the IRSM

To reveal the molecular basis of the IRSM, we first compared the gene expression profiles of the IRSM-related genes. We observed that all of them showed lower expression in high-risk vs. low-risk cases (Figure 5A). In addition, we examined the expression profiles of genes within the yellow module. We found that 89.92% (339/377) of the yellow module genes displayed reduced expression patterns in the high-risk group compared to the low-risk group (Figure 5B, Wilcoxon test *p* < 0.05). All these results suggested that distinctive molecular patterns existed between the different-risk-score groups, and the lower expression of key genes induced the expression of downstream genes and subsequently promoted EC tumorigenesis and development.

### 3.6. Clinical and Molecular Characteristics of Different IRSM Groups

To explore the relationship between the IRSM and clinical characteristics, we first evaluated whether the IRSM could serve as an independent predictor for EC patient survival. Multivariate Cox regression analysis was applied to estimate the relationship between the IRSM and clinical characteristics such as age, FIGO grade, FIGO stage and pathological subtype. The results showed that the IRSM was an independent predictor of overall survival (Figure 6A). Meanwhile, patients in the low-risk group presented fewer deaths or a less severe illness status (Figure 5B), which is consistent with the previous result that patients in the low-risk group had better clinical outcomes.

Considering that Microsatellite Instability (MSI) and Tumor Mutation Burden (TMB) play vital roles in EC, we compared the MSI-H status and TMB between the high- and the low-risk groups. We found that the patients in the low-risk group presented significantly higher MSI-H status (*p*-value = 0.001855) and TMB (*p*-value = 7.6 × 10^−10^) than the patients in the high-risk group (Figure 6B and Table 2). Subsequently, we further compared the mutation frequency of mismatch repair (MMR) genes between the two groups. A higher mutation rate of MMR genes was observed for patients in the low-risk group than for those in the high-risk group (Figure 6C,D and Table 2). All these results indicated that the low-risk group was associated with an immune-hot phenotype, and the patients in this group might benefit from immunotherapy.

Next, we further compared several important molecular characteristics such as *POLE* and *TP53* mutation and the expression patterns of immune checkpoint genes as well as human leukocyte antigen (HLA) genes. A higher mutation rate of *POLE* (*p*-value = 0.0006069) and a lower mutation rate (*p*-value = 2.195 × 10^−5^) of *TP53* were observed in the low-risk group (Table 2), indicating the low-risk group had a favorable prognosis. In addition, over-expression patterns of immune checkpoint genes and HLA genes were observed in the low-risk group (Figure 6E,F), suggesting that the patients in the low-risk group were more likely to benefit from immunotherapy.

### 3.7. TME Immune Infiltration Characteristics of the Different IRSM Groups

Since the IRSM was constructed based on immune-related key genes, we further compared the tumor immune microenvironment between the two groups. Particularly, we dissected the relative abundance of 22 types of immune cells by the CIBERSORT algorithm. We observed that 59.09% (13/22) of immune cell types showed significant differences in cell infiltration in the high- and low-risk groups. Particularly, the relative proportions of resting dendritic cells, M1 macrophages, resting mast cells, plasma cells, activated CD4 memory T cells, CD8 T cell, T follicular helper cells and regulatory T cells were higher in the low-risk group, while the relative proportions of activated dendritic cells, M0 macrophages, activated mast cells, resting CD4 memory T cells, CD4 naïve T cells were higher in the high-risk group (Figure 7).

### 3.8. The IRSM Is Associated with Immunotherapy Response in EC Patients

To check the relationship of the IRSM with immunotherapy, we predicted the response to immunotherapy using the Tumor Immune Dysfunction and Exclusion (TIDE) tool which has been widely used as an indicator of immunotherapy response. Meanwhile, we calculated the TIDE score of each patient; patients with lower TIDE scores might benefit from an anti-PDL1/CTLA4 treatment. After examining the response to immunotherapy, we noticed that 91.25% (73/80) of the patients who responded to immunotherapy were primarily in the low-risk group (Figure 5B). Then, we compared the TIDE scores of the high- and low-risk groups. The TIDE score of the low-risk group was significantly lower than that of the high-risk group (*p*-value = 2.2 × 10^−16^, Figure 8A), indicating that the patients in the low-risk group were more likely to respond to immunotherapy. In short, our results highlighted that the IRSM has a good capability to stratify patients, identifying those who might benefit from immunotherapy.

### 3.9. The IRSM Is Associated with Chemotherapy Response in EC Patients

To check the relationship of the IRSM with chemotherapy, we computed the half-maximal inhibitory concentration (IC50) of seven commonly used chemotherapeutic drugs (Cisplatin, Cyclophosphamide, Docetaxel, Fluorouracil, Paclitaxel, Tamoxifen and Topotecan). The therapy responses to the above-mentioned chemotherapeutic drugs in the low-risk group were significantly better than those in the high-risk group (Figure 8B), suggesting that patients in the low-risk group were more sensitive to chemotherapy.

### 3.10. Combining Immunotherapy with Chemotherapy Could Enhance the Treatment Effects

Given the fact that chemotherapy combined with immunotherapy shows better efficacy than either treatment alone, we further investigated the relationships of the IRSM with combined immunotherapy and chemotherapy. According to previous predictions, most patients responding to immunotherapy were concentrated in the low-risk group. Thus, we focused our further analysis on the low-risk group. We sub-grouped the patients in the low-risk group into an immunotherapy-responsive group and an immunotherapy-nonresponsive group. Subsequently, we compared the sensitivity to the above-mentioned chemotherapeutic drugs of the two groups. For each chemotherapeutic drug, the patients in the immunotherapy-responsive group had a better chemotherapy response than the patients in the immunotherapy-nonresponsive group (Figure 8C). This result indicated that immunotherapy coupled with chemotherapy could enhance the treatment effect in patients.

## 4. Discussion

Endometrial cancer (EC) is the most common gynecologic cancer worldwide. The overall survival of EC patients has improved over the past decade but remains unsatisfying due to a lack of reliable screening. In addition, even though immunotherapy has been used in the clinic, the screening of patients responding to immunotherapy remains challenging. Therefore, there is a strong need to develop an effective approach to optimize the treatment screening and improve patient survival.

In this study, we applied the WGCNA algorithm to mine immune-related modules and key genes. We found that the yellow module showed the closest relationship with immune cell infiltration. Meanwhile, the Go enrichment analysis confirmed that the yellow module was an immune-related module. In addition, seven key genes of the yellow module were identified, and five of them (*CXCR3*, *CD3D*, *CCL5*, *CD3E* and *IL2RG*) were screened to construct an immune-related risk score model (IRSM). *CXCR3* codes for a chemokine receptor and is primarily expressed in CD4+ and CD8+ T cells. *CXCR3* plays an important role in T cell trafficking during inflammation [42,43]. *CD3D* and *CD3E* are involved in the encoding of the T cell receptor/CD3 complex (TCR/CD3 complex), which is a cell surface structure and plays a crucial role in antigen recognition and T cell activation [44,45]. *CCL5* is a member of the C–C chemokine family, and its overexpression was linked with CD8+ T cell infiltration in solid tumors [46]. Meanwhile, *CCL5* promotes the recruiting of various leukocytes into inflammatory sites [47,48]. *IL2RG* encodes the cytokine receptor γ chain and is an important component of many interleukin receptors such as *IL-2*, *IL-4*, *IL-7*, *IL-9*, *IL-15* and *IL-21* [49]. All these pieces of evidence supported the conclusion that the IRSM-related key genes are immune-related genes and play critical roles in tumorigenesis and development.

The immune-related risk score model (IRSM), a novel screening approach for EC treatment selection was established. Particularly, the IRSM was found to be an independent predictor of poor prognosis. All IRSM-related genes were more highly expressed in the low-risk group. Univariate Cox regression analysis also indicated that the IRSM-related genes were associated with good prognosis. This was consistent with the expression pattern of these genes within the two groups. In addition, the Human Protein Atlas proved that the IRSM-related genes were highly associated with good prognosis [50,51]. Their analysis not only confirmed our results but also highlighted the important roles of these five genes in cancer progression. Next, we further examined the expression level of the yellow module genes within the two groups. Most of the yellow module genes presented over-expression in the low-risk group. Therefore, we suggest that alterations of immune-related key genes affect the expression pattern of downstream co-expressed genes and further influence EC prognosis.

Considering that microsatellite instability/deficient mismatch repair (MSI-H/dMMR), tumor mutation burden (TMB), *POLE* and *TP53* mutation play vital roles in EC, their relationships with IRSM were investigated. We observed that the MSI-H status, the mutation rate of MMR genes as well as of *POLE* and TMB were significantly higher in the low-risk group than in the high-risk group. Alicia et al. highlighted that higher MSI-H/dMMR could produce a better clinical response (improve the OS) [52,53,54]. In addition, a large number of clinical studies have demonstrated that MSI-H/dMMR serves as an effective indication for cancer immunotherapy [55,56,57], and patients with high MSI-H/dMMR are more likely to benefit from immunotherapy. Robert M. et al. stated that a high value of TMB is associated good prognosis [30,31,32]. Meanwhile, TMB also serves as a biomarker of immunotherapy, and a higher TMB is linked to a better immune response. Melissa K. et al. observed that patients with *POLE* mutations usually have a good prognosis [58,59,60]. The *POLE* mutational status is one of the most important prognostic biomarkers in EC. Patients with somatic *POLE* mutations present a favorable prognosis. Furthermore, *POLE* mutation could regulate the immune response and ha been treated as a biomarker for immunotherapy response [61]. As for *TP53* mutations, we observed that the low-risk group presented a lower mutation frequency than the high-risk group. Definitely, the *TP53* status was proven as an independent prognostic biomarker for EC patients. Several studies [62,63,64,65,66,67] demonstrated that *TP53* mutation is associated with poor clinical outcomes. All these pieces of evidence are consistent with our analysis, proving the prognosis prediction ability of the IRSM and its important role in cancer immunotherapy. Specially, considering the molecular classification of EC is based on the *POLE* and *TP53* gene status and microsatellite status, we regard that the IRSM may be related to EC molecular classification.

Since immune checkpoint blockade therapy has become a crucial weapon in the fight against cancer, the relationship between the IRSM with immunotherapy was estimated. We first predicted the immunotherapy response of each patient and then compared the responses in the high- and low-risk groups. We noticed that most of the patients responding to immunotherapy were concentrated in the low-risk group. We further compared the expression patterns of immune checkpoint genes (ICGs) between the two groups and observed over-expression patterns of ICGs in the low- vs. the high-risk group. High expression of PD-L1 is associated with an effective response to anti-PD-1/anti-PD-L1 therapy [68,69]. In addition, we also compared the expression profiles of HLA genes between the high- and the low-risk groups. Increased expression of HLA genes was noticed in the low-risk group. Evelien Schaafsma et al. [70] proved that HLA gene expression is positively related to a patient’s immune checkpoint blockade response. The immune infiltrating cell contents in the high- and the low-risk groups were compared, and a more immunologically active TME (also called immune-hot TME) was observed in the low-risk group. We also examined the expression pattern of the IRSM genes (*CXCR3, CD3D, CCL5, CD3E* and *IL2RG*) within the two groups. All IRSM-related genes presented higher expression in the low-risk group. Chheda et al. demonstrated that the induction of *CXCR3* enhanced the migration of T cells to tumors and promoted the therapeutic effect of anti-PDL1 therapy [71]. *CD3E* and *CD3D* play important roles in the positive regulation of T-cell activation and leukocyte cell–cell adhesion and are considered to be the main determinants of tumor immunotherapy efficacy [45,72]. High expression of *CD3E* is associated with anti-PD1 immunotherapy efficacy [73]. *CD3D* has been reported as a potential biomarker for immunotherapy [74,75,76]. *CCL5* could promote an anti-tumor response by recruiting anti-tumor immune cells to the TME, which enhance the immunotherapy response [46,77,78,79,80]. *IL2RG* is an important component of the γ c family of cytokines, including IL-2. IL-2 was proven to be an effective immunotherapy by the FDA [81,82,83,84]. Taken together, all the above observations confirmed the close relationship of the IRSM with immunotherapy.

The relationship of the risk score model with chemotherapy was evaluated. We observed that patients in the low-risk group were more sensitive to chemotherapeutic drugs. In addition, we also noted that patients responsive to immunotherapy were also more sensitive to chemotherapy. Previous studies pointed out that patients who responded to immunotherapy were more sensitive to chemotherapy, implying that immunotherapy could enhance the sensitivity of tumor cells to chemotherapeutic drugs [85,86]. In short, all the above observations confirmed the close relationship of the IRSM with chemotherapy and proved the enhancement immunotherapy efficacy when combined with chemotherapy in the treatment of EC.

Despite the extensive observations and consistent results generated by our analysis, some limitations of this study should be noticed. Firstly, even though the performance of the IRSM in prognostic prediction was verified, to facilitate the widespread clinical application of the IRSM, its ability to predict patient survival still needs to be confirmed using additional independent datasets. Secondly, since the patients’ response to immunotherapy or chemotherapy was predicted based on bioinformatics tools, analysis of patients receiving immunotherapy or chemotherapy is still needed to verify the relationship of the IRSM with immunotherapy and chemotherapy. Thirdly, despite evaluating the relationship of the IRSM with immune infiltration, how immune infiltration works in tumor microenvironments remains unclear, and the corresponding mechanisms need to be experimentally verified in the future. Last but not least, while the molecular basis as well as the biological properties of these immune-related key genes were inferred, their mechanisms of action remain elusive, and further experimental validations are still needed.

## 5. Conclusions

In conclusion, we constructed an immune-related risk score model (IRSM) for endometrial cancer. The IRSM was associated with poor prognosis and could serve as an independent survival predictor. The IRSM also showed a high predictive value for immunotherapy and chemotherapy response. Meanwhile, the clinical and molecular characteristics confirmed the predictive effect of the IRSM for prognosis and the response to immunotherapy and chemotherapy. Moreover, patients responding to immunotherapy were also more sensitive to chemotherapy. Our analysis not only will help optimize personalized treatments but also provides a deeper understanding of endometrial cancer progression.

## Figures and Tables

**Figure 1 cancers-15-03673-f001:**
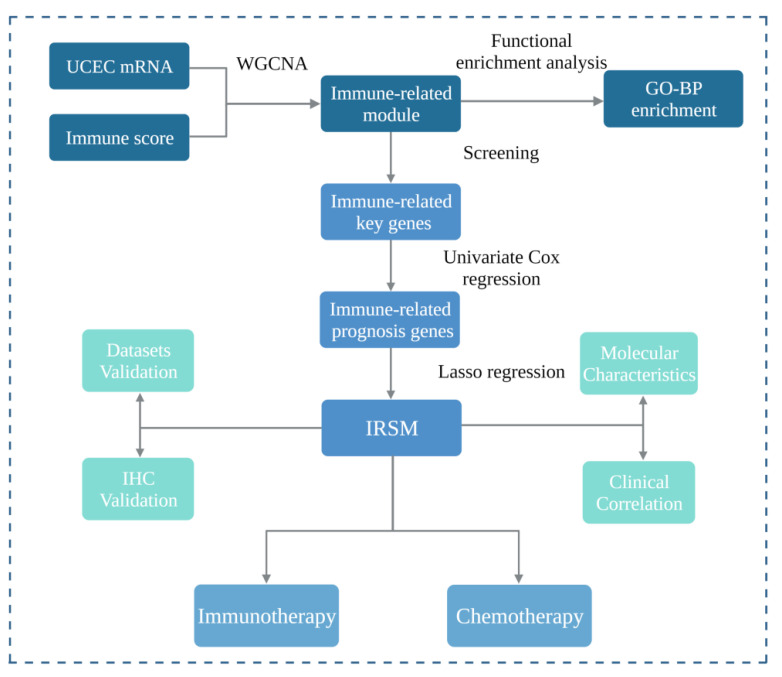
Study workflow. Firstly, we applied weighted gene co-expression network analysis (WGCNA) to construct a gene co-expression network and further mined the immune-related module. Functional enrichment analysis was performed, and immune-related key genes were obtained from the immune-related module. Secondly, we selected immune-related prognosis genes by univariate Cox regression and then constructed an immune-related risk score model (IRSM) using the Lasso regression in the training dataset. Thirdly, we estimated the performance of the IRSM in prognosis prediction. Meanwhile, the roles of the IRSM genes in EC development were validated by immunohistochemistry. Fourthly, we evaluated the relationship between the IRSM and EC molecular and clinical characteristics. Fifthly, we assessed the relationships between the IRSM and both immunotherapy response and chemotherapy sensitivity.

**Figure 2 cancers-15-03673-f002:**
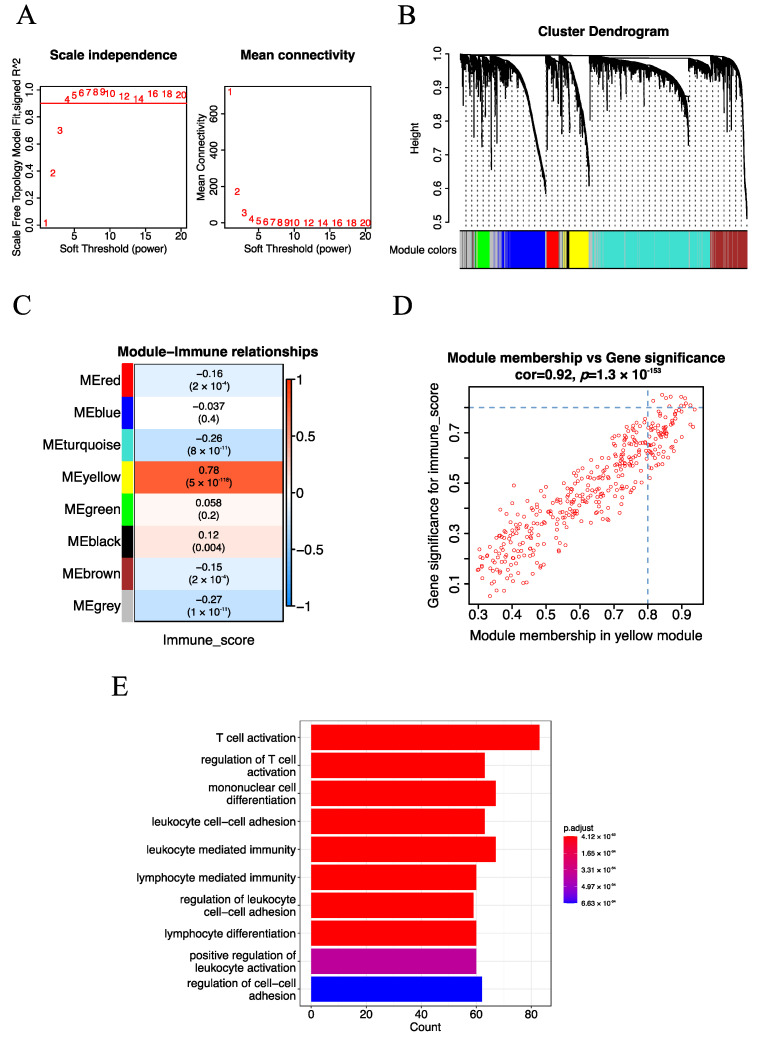
Identification of the immune-related module and key genes for EC. (**A**) The screening of soft-thresholding power (β). (**B**) Cluster dendrogram of the co-expression network modules (1-TOM). (**C**) Heatmap of the correlation of consensus modules with immune traits. The row represents the module eigengenes (MEs), and the column represents the immune score. Pearson correlation coefficient (r) as well as *p*-value are shown in the cells. (**D**) Scatter plot analysis of the yellow module. The key genes were screened out in the upper-right area, where GS > 0.8 and MM > 0.8 (TOM, topological overlap matrix. GS, gene significance. MM, module membership). (**E**) The top 10 significantly enriched biological processes of the yellow module.

**Figure 3 cancers-15-03673-f003:**
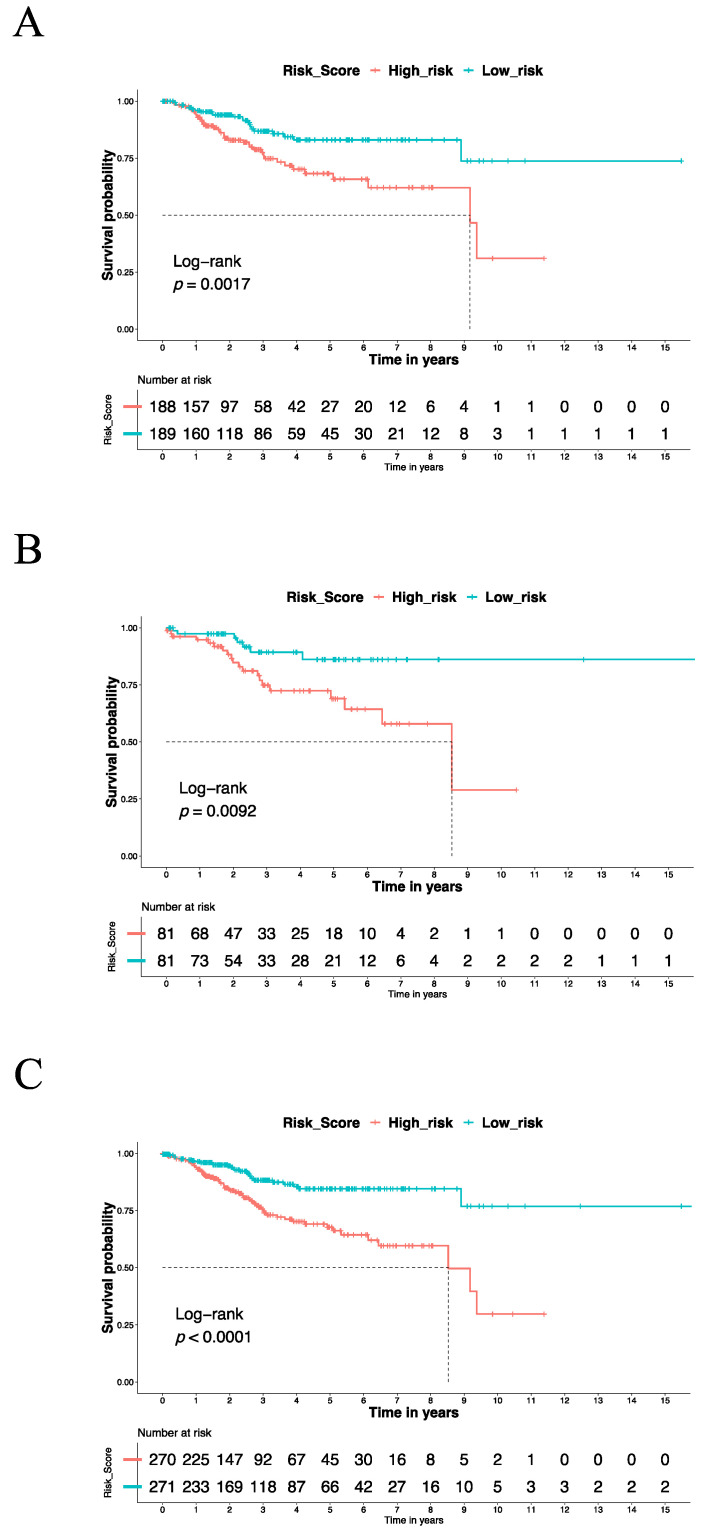
The Kaplan–Meier estimator was used to estimate the survival of high-risk vs. low-risk EC patients based on the IRSM. (**A**) The OS for the training dataset. (**B**) The OS for the validation dataset. (**C**) The OS for the whole dataset.

**Figure 4 cancers-15-03673-f004:**
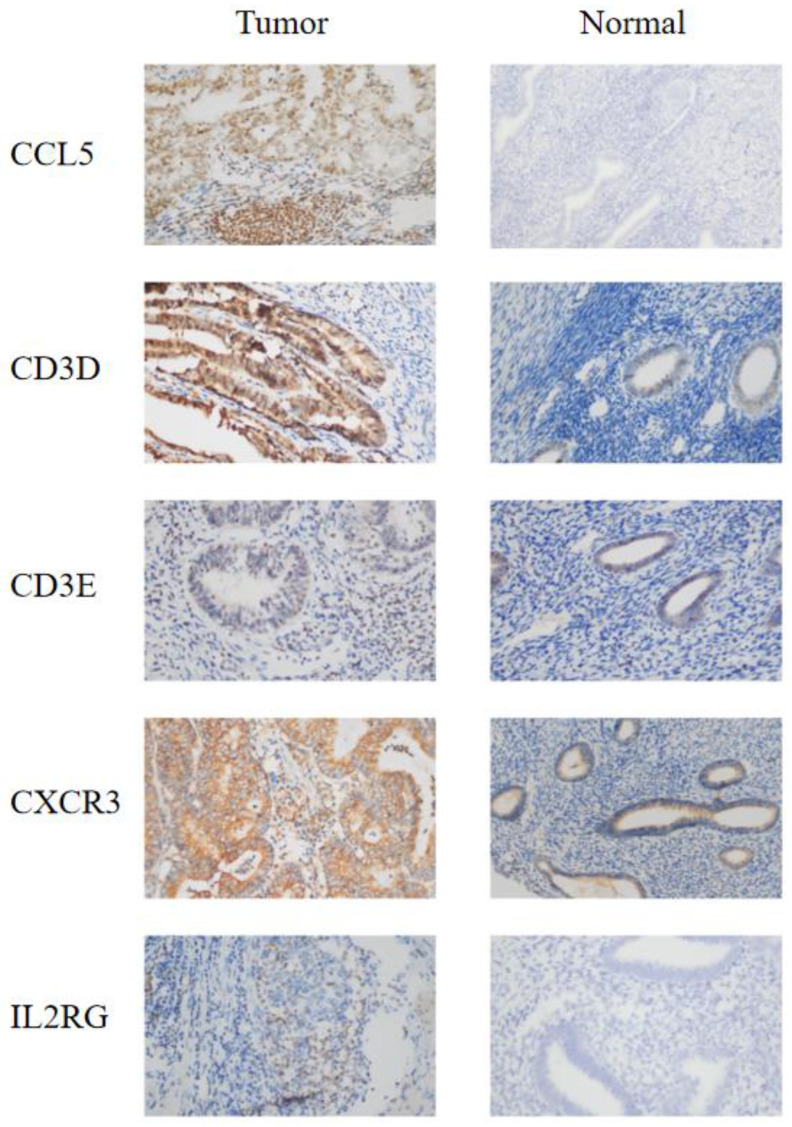
IHC staining of the model genes in EC tumor tissue and normal tissue. (**left**) Endometrial carcinoma tumor tissue; (**right**) normal tissue. The microscope magnification is 40×.

**Figure 5 cancers-15-03673-f005:**
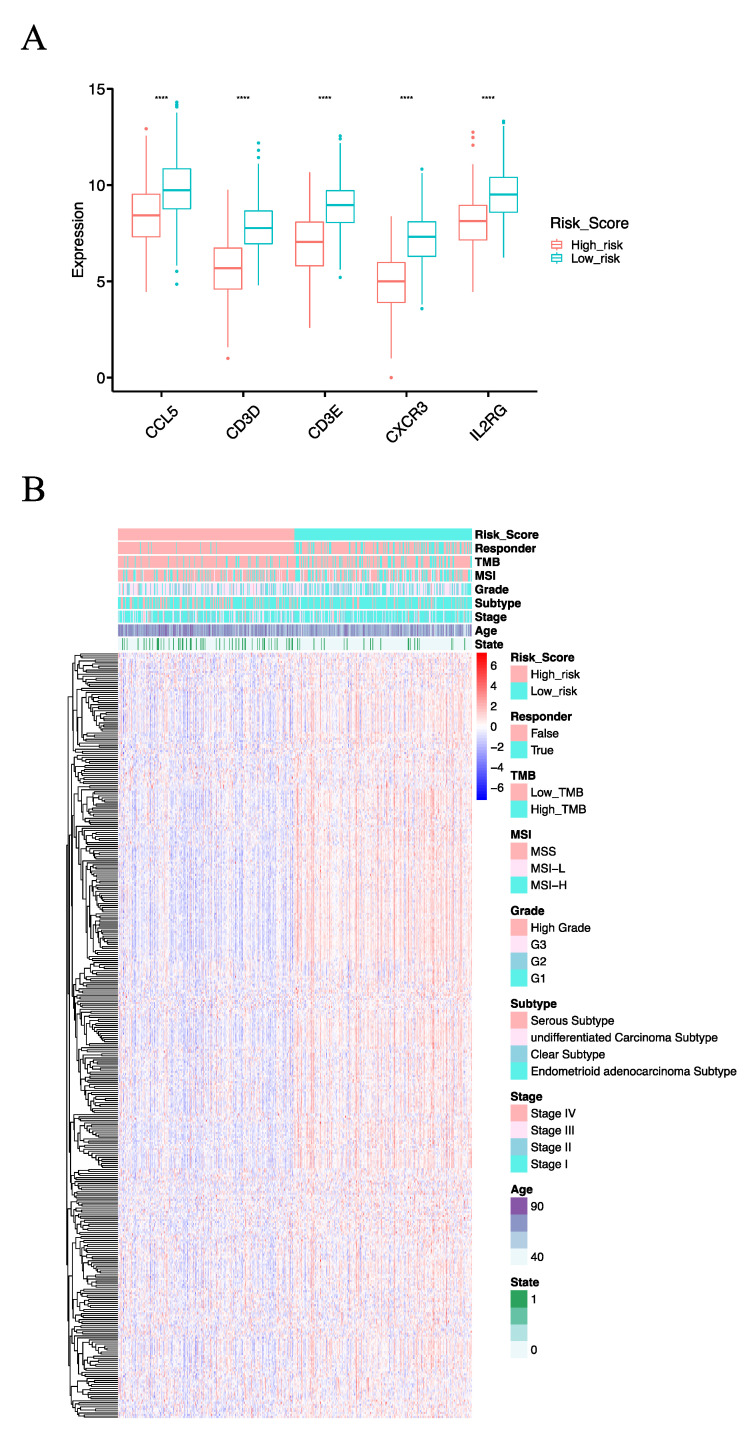
The molecular basis of the IRSM. (**A**) Box plot comparing the expression pattern of model genes in the high- and low-risk groups. (**B**) Landscape of the high- and low-risk groups. (Notes: ****: *p*-value < 0.0001).

**Figure 6 cancers-15-03673-f006:**
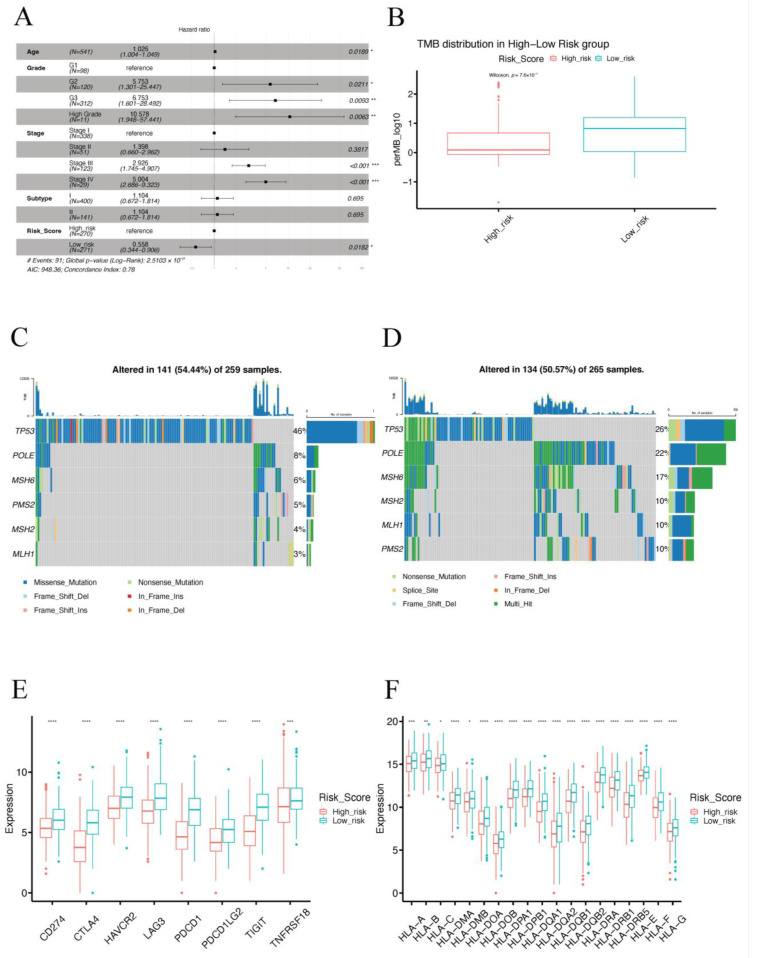
Clinical and molecular characteristics of different IRSM groups. (**A**) Multivariate Cox regression analysis. (**B**) Box plot comparing the TMB in the high- and low-risk groups. (**C**) Mutation visualization of TP53, POLE and MMR genes in the high-risk group. (**D**) Mutation visualization of TP53, POLE and MMR genes in the low-risk group. (**E**) Box plot comparing the expression pattern of immune checkpoint genes between the high-risk and the low-risk groups. (**F**) Box plot comparing the expression pattern of HLA genes in the high-risk and the low-risk groups. (Notes: * indicates *p*-value < 0.05; ** indicates *p*-value < 0.01; ***: *p*-value < 0.001; ****: *p*-value < 0.0001).

**Figure 7 cancers-15-03673-f007:**
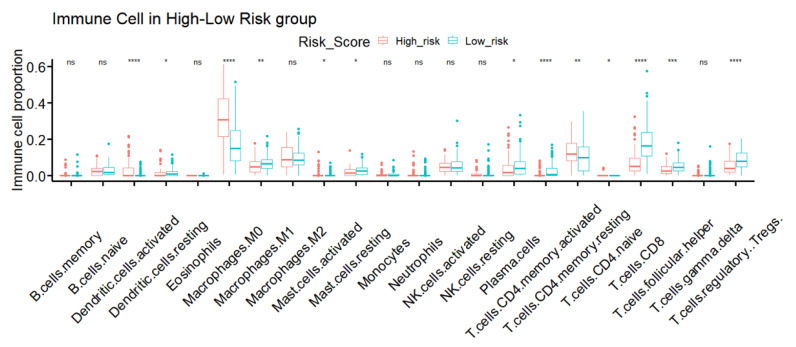
TME immune infiltration characteristics of different IRSM groups. (Notes: * indicates *p*-value < 0.05; ** indicates *p*-value < 0.01; ***: *p*-value < 0.001; ****: *p*-value < 0.0001; ns: not significantly).

**Figure 8 cancers-15-03673-f008:**
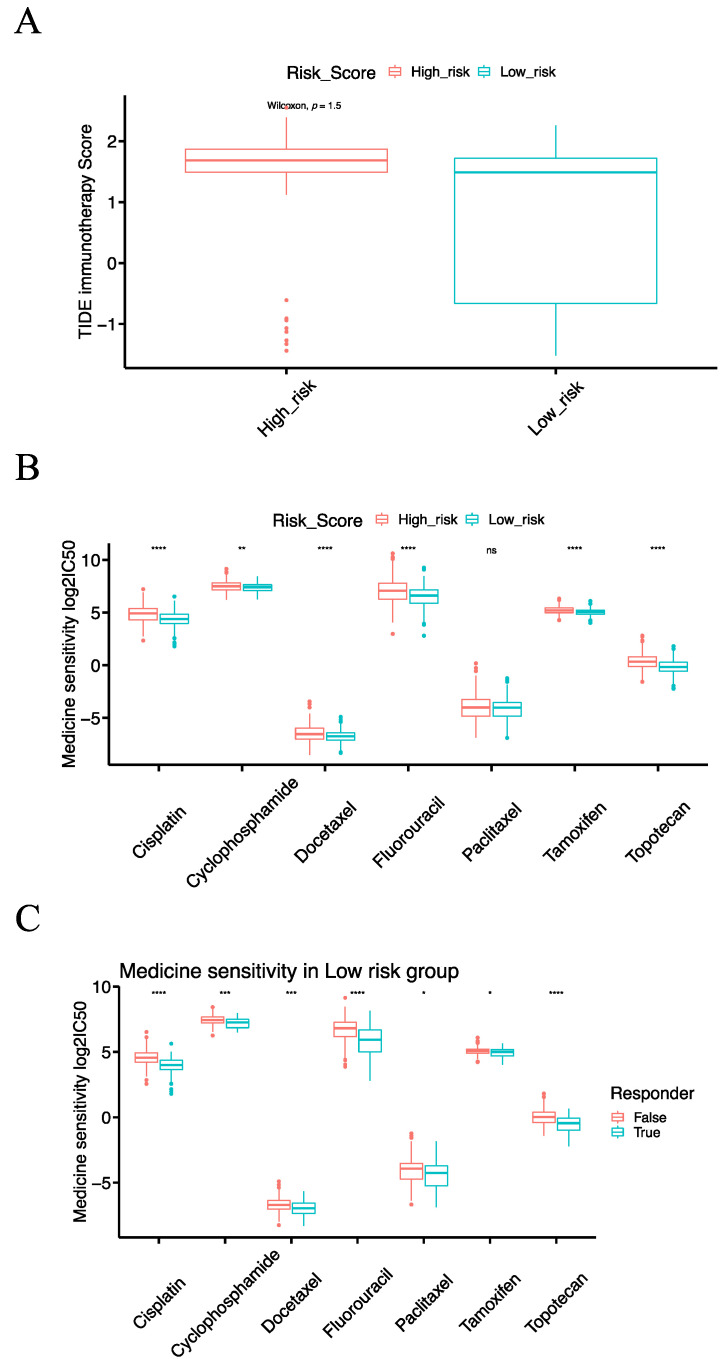
The relationship of the risk score model with immunotherapy and chemotherapy. (**A**) Box plot comparing the TIDE score of immunotherapy response between the high-risk and the low-risk groups. (**B**) Box plot comparing the IC50 values of seven commonly used chemotherapy drugs between the high-risk and the low-risk groups. (**C**) Box plot comparing the IC50 values of seven commonly used chemotherapy drugs for patients in the low-risk group with or without immunotherapy response. (Notes: * indicates *p*-value < 0.05; ** indicates *p*-value < 0.01; ***: *p*-value < 0.001; ****: *p*-value < 0.0001; ns: not significantly).

**Table 1 cancers-15-03673-t001:** Demographic and clinical characteristics.

Characteristics	Sample Size
Patient No.	541
Pathological subtype No.	
Endometrioid adenocarcinoma subtype	400
Serous subtype	138
undifferentiated carcinoma subtype	2
Clear subtype	1
FIGO Stage	
Stage I	338
Stage II	51
Stage III	123
Stage IV	29
FIGO Grade	
G1	98
G2	120
G3	312
High grade	11
Age (years) *	
Range	31~90
Median	64
Follow-up (days)	
Range	0~6859
Median	902
Status	
Alive	450
Dead	91
MSI	
MSI-H	157
MSI-L	43
MSS	297
TMB	
High-TMB	105
Low-TMB	419

* Some information is missing for certain patients.

**Table 2 cancers-15-03673-t002:** The comparison of clinical and molecular characteristics in the high- and low-risk groups.

Groups	Statistical Method	*p*-Value
MSI in high- and low-risk	Chi-square test	0.001855
TMB in high- and low-risk	Chi-square test	1.012 × 10^−6^
TIDE in high- and low-risk	Chi-square test	<2.2 × 10^−16^
TP53 in high- and low-risk	Chi-square test	2.195 × 10^−5^
POLE in high- and low-risk	Chi-square test	0.0006069
MSH6 in high- and low-risk	Chi-square test	0.0006415
MSH2 in high- and low-risk	Chi-square test	0.03334
MLH1 in high- and low-risk	Chi-square test	0.002758
PMS2 in high- and low-risk	Chi-square test	0.3518

## Data Availability

The public dataset and The First Affiliated Hospital of Chongqing Medical University dataset were used in this study. The public dataset is from the TCGA-UCEC project and is available in GDC TCGA Data Portal (https://portal.gdc.cancer.gov/, accessed on 13 January 2022) and the “TCGABiolinks” R package. The First Affiliated Hospital of Chongqing Medical University was a control dataset.

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
