# Peer review of "Prediction of Prognosis, Immunotherapy and Chemotherapy with an Immune-Related Risk Score Model for Endometrial Cancer"

_cancers, 2023, doi:10.3390/cancers15143673_

Round 1

Reviewer 1 Report

This manuscript proposed a novel model for predicting the prognosis, immunotherapy response, and chemotherapy sensitivity of endometrial cancer. I have several questions:

1.     The figure legend of Figure 1 was not provided. Please add it.

2. In the methods, ' Functional enrichment analysis' section, “…. Only GO terms with q-value less than 0.05 were considered significantly enriched and the top 10 enriched GO items were displayed using the “Graphics” R package”. Also, in the 'Construction and validation of the risk score model' section, “…To construct a risk score model, for each immune-related key gene, we first applied univariate Cox regression analysis to evaluate the relationship with patient survival outcome by “survival” R package”. The software that was used in your research should be cited.

3. The immune checkpoint genes and human leukocyte antigen genes have been introduced in the Method section. It’s not necessary to repeat them again in the Results section.

4. According to the Results “…In addition, we performed GO enrichment analysis to explore the biological functions of the yellow module. We observed that immune-related biological processes such as…”, it seems only biological process analysis was performed during Go enrichment analysis. This should be pointed out in the method ' Functional enrichment analysis' section.

5. Overall, the manuscript was well written. However, some Typos and possible grammar errors still exist. Please generally check the grammar and language.

Some Typos and possible grammar errors still exist. Please generally check the grammar and language.

Reviewer 2 Report

This study aims to construct an immune-related risk score model (IRSM) that is derived from survival lasso cox models, which could be used to predict the prognosis of patients. The IRSM score could be linked to immunotherapy and chemotherapy out of endometrial cancer. The featured genes used in IRSM were validated with IHC. The molecular basis and clinical characteristics of the risk score model were also revealed. Overall, this is an interesting and solid work. The followings are my comments.

Comment 1: In subsection 2.5. Functional enrichment analysis, “…The hypergeometric test was used to calculate p-values for gene set enrichment and the false discovery rate (BH FDR) was used to calculate q value for multiple test compensation”, a multiple testing correction with BH FDR. BH should be spelled out.

Comment 2: In subsection 3.2. Construction of IRSM based on immune-related key genes, the description of Figure S3 is not quite clear, please pointed out the subfigure in the proper text.

Comment 3: The figures resolution is not good enough. Please provide some higher resolution figures.

Comment 4: There are still some mistakes in grammar and language, the authors should carefully check this manuscript.

Comment 5: The immune-related risk score model (IRSM) is constructed purely with the gene effect size and expression obtained from the previous lasso Cox survival models. Did the gene expression get normalized to avoid potential bias in the IRSM model.

many obvious typos observed, such as Fig 1 `screeing`, in Abstract: `Backgroud`. The authors need to have extensive efforts in improving the writing. 

Round 2

Reviewer 1 Report

The authors have answered my questions.